# Easyreporting simplifies the implementation of Reproducible Research layers in R software

Dario Righelli [1,2]*, Claudia Angelini[2]*

**1** Department of Statistical Sciences, University of Padova, Padua, Italy, **2** Istituto per le Applicazioni del Calcolo "Mauro Picone", National Research Council, Naples, Italy

\* d.righelli@na.iac.cnr.it (DR); c.angelini@iac.cnr.it (CA)

## Abstract

During last years "irreproducibility" became a general problem in omics data analysis due to the use of sophisticated and poorly described computational procedures. For avoiding misleading results, it is necessary to inspect and reproduce the entire data analysis as a unified product. Reproducible Research (RR) provides general guidelines for public access to the analytic data and related analysis code combined with natural language documentation, allowing third-parties to reproduce the findings. We developed *easyreporting*, a novel R/Bioconductor package, to facilitate the implementation of an RR layer inside reports/tools. We describe the main functionalities and illustrate the organization of an analysis report using a typical case study concerning the analysis of RNA-seq data. Then, we show how to use *easyreporting* in other projects to trace R functions automatically. This latter feature helps developers to implement procedures that automatically keep track of the analysis steps. *Easyreporting* can be useful in supporting the reproducibility of any data analysis project and shows great advantages for the implementation of R packages and GUIs. It turns out to be very helpful in bioinformatics, where the complexity of the analyses makes it extremely difficult to trace all the steps and parameters used in the study.

## Introduction

Due to accidental mistakes or misuse of sophisticated computational methods, many research findings in omics science are considered false (or partially incorrect) [1]. Moreover, in several cases, published results are not entirely reproducible due to the lack of information. For example, the analysis of the massive amount of omics data produced by high-throughput technologies requires combining several different methodologies from the preprocessing, data cleaning, and normalization to the downstream analysis. Therefore, it becomes challenging to trace all the steps and the parameters used within a complete analysis. Consequently, the lack of details, such as user-parameter or subtle data manipulation made with small code lines not reported in the material and methods sections of a manuscript, can lead to findings that are not reproducible. To prevent misleading results, several authors suggested adopting some best practices [2–4] that should help in publishing reproducible results. Nevertheless, the proposed approaches can be time-consuming and require significant effort by researchers. Therefore, to

**Data Availability Statement:** All Supplementary Files and their source codes are available at https://github.com/drighelli/easyreporting_supplementary.

**Funding:** The work has been partially supported by the Regione Campania Project ADViSE assigned to

Dr. Claudia Angelini. The funders had no role in study design, data collection and analysis, decision to publish, or preparation of the manuscript. There was no additional external funding received for this study.

**Competing interests:** The authors have declared that no competing interests exist.

fully exploit the advantages of Reproducible Research (RR), it is still necessary to provide tools that can trace all the details using automatic procedures [5, 6].

Recently, the scientific community proposed several approaches to support RR by developing tools that require a lower cost in terms of time and efforts to be used [5–9]. Among the different approaches, one common idea is to describe the steps with an analysis report built up as a mixture of natural language sentences along with computational language and graphical outputs. This document should include: i) the analyzed data, ii) the Code Chunks (CCs), iii) results and intermediate outputs (as tables and figures), and iv) all information that can enhance the work comprehensibility and reproducibility. Using human-readable reports instead of other procedures (for example, virtualization solutions such as docker containers) have the advantage that the final document can be easily understood by non-expert users, whereas docker containers require computationally experienced users. Moreover, a human-readable report can be enriched with comments and favors knowledge transfer. Nevertheless, the two approaches are complementary and can be combined to achieve full reproducibility in terms of input/output of each algorithm/function and the possibility to re-create a computational environment that does not depend on specific user installations.

The R community proposed several solutions based on the literate statistical programming, like *sweave* [10], *knitr* and *rmarkdown* [11]. Within this framework, the authors can release a data analysis as a human-readable document that incorporates data, computational methods (including the short lines of code that are often omitted in a high-level description of the computational procedure), user-parameters, tables, and figures. Moreover, this report is automatically updated each time the analyst introduces some workflow changes to preserve complete reproducibility. R Studio (https://rstudio.com) already contains several functionalities that can help an analyst compiling detailed reports. Other R packages, such as Drake [12], go through the same directions.

Even though *rmarkdown* is very popular and easy to use inside the R community for writing step-by-step analysis reports, its usability when developing automated tools as Graphical User Interfaces (GUI) or packages is limited. Despite several efforts, incorporating a RR layer in other software and automatically tracing all the steps performed during a point-and-click analysis is still challenging. In the past, we proposed a solution with the RNASeqGUI [13, 14] project. RNASeqGUI is a GUI for analyzing RNA-seq data that automatically traces the analysis steps and reports them in a unique report. Although very useful, this solution did not allow the user to add personal comments, a particularly relevant requisite for knowledge transfer. Moreover, its implementation was time-consuming.

In light of these reasons, we developed *easyreporting*, a R/Bioconductor package allowing us to construct reports in different formats (i.e. HTML, PDF) that automatically incorporate comments with data, code, plots, and tables. In this work, we describe the *easyreporting* class and its methods. Then, we show i) how *easyreporting* can be used to generate user analysis reports and ii) how *easyreporting* can be used to implement packages and GUIs that automatically trace their functions and produce an analysis report. This latter feature makes *easyreporting* a particularly relevant and practical tool to improve user-friendly software. Moreover, the same approach can be used to create novel R packages or to trace user-defined functions.

## Materials and methods

### Implementation

*Easyreporting* is an open-source R/Bioconductor package aimed to 1) support analysts to speed up the compilation of their analysis reports and 2) help developers to integrate a RR

layer inside their R software products (such as GUIs and packages). While the first aim can also be easily achieved using other similar tools, the latter constitutes one of the main advantages of our solution. In such a way, thanks to minimal efforts on the developers' side, the end-user can obtain an *rmarkdown* file that incorporates the source code generated during the analysis with the user-friendly tools. Once compiled, this document can then be published as supplementary material of a scientific article, helping the interested community to reproduce the computational part of the work entirely, as suggested in [15]. Moreover, the document can be easily organized into sections, describing different analysis steps, and enriched with natural language comments, making the report more explainable to increase the knowledge transfer.

## General description and initialization

*Easyreporting* is structured as an S4 class representing a schematic view of a *rmarkdown* file (see Fig 1). Thanks to *easyreporting*, an analysis report can be seen as a particular instance of the package class, where the attributes represent the report characteristics. Within this class, the available methods are useful for attribute manipulations and for inserting comments and organizing section titles inside the report.

When *easyreporting* is used to create a report for novel analysis, the analyst needs to initialize an instance of the *easyreporting* class with the *easyreporting()* constructor function, passing as mandatory arguments the path and the name of the report file accompanied by its title. Optionally, it is possible to specify one or more authors' names (with emails, affiliations, affiliation websites, ORCIDs, and personal websites, like in a publication). Additionally, during the class creation, it is possible to define a bibliography latex file through the *bibliography* argument, that will be compiled as reference list at the end of the report (see Table 1).

In this way, each analysis/project is uniquely associated with a specific *easyreporting* instance, and hence to the corresponding *rmarkdown* file. The initialization step is transparent to any software user since the developers handle the tool's back-end.

During the initialization, the class constructor automatically creates the report file inside the specified folder tree, setting up its header and declaring the general options for the *rmarkdown* file. As soon as the analyst or the user proceeds with his analysis, the *rmarkdown* file is updated with a new CC each time the analysis software performs a new analysis step. When the analysis is complete, it is possible to compile the report using the *compile()* method, which produces the final report (in the user-defined format) and appends a final CC with the *sessionInfo* to trace all the packages versions used for the analysis. Additionally, this process creates two others optional sections, one with the cited references when a bibliography file has been specified during the easyreporting instance creation, and another named *Resources Availability* with the resources specified with the *addResource()* method (see Section 3.9 in S1 File for more details).

## General exploitation

The *easyreporting* class is equipped with several methods for *rmarkdown* CC construction (see Table 1 for the full list). Once an *easyreporting* instance is available, it is possible to organize the report by inserting up to six levels of titles by using the *mkdTitle* method. It is also possible to add natural language comments with *mkdGeneralMsg*. The latter feature is particularly relevant to make the analysis more understandable.

For the implementation of the CCs creation, we suggest two main approaches based on the methods available in the class (see the examples in Listings 2 and 3 shown in the Results Section): i) The first approach builds a CC as a typical step-by-step process. It consists of opening a CC (*mkdCodeChunkSt*), adding variables assignments and/or function callings

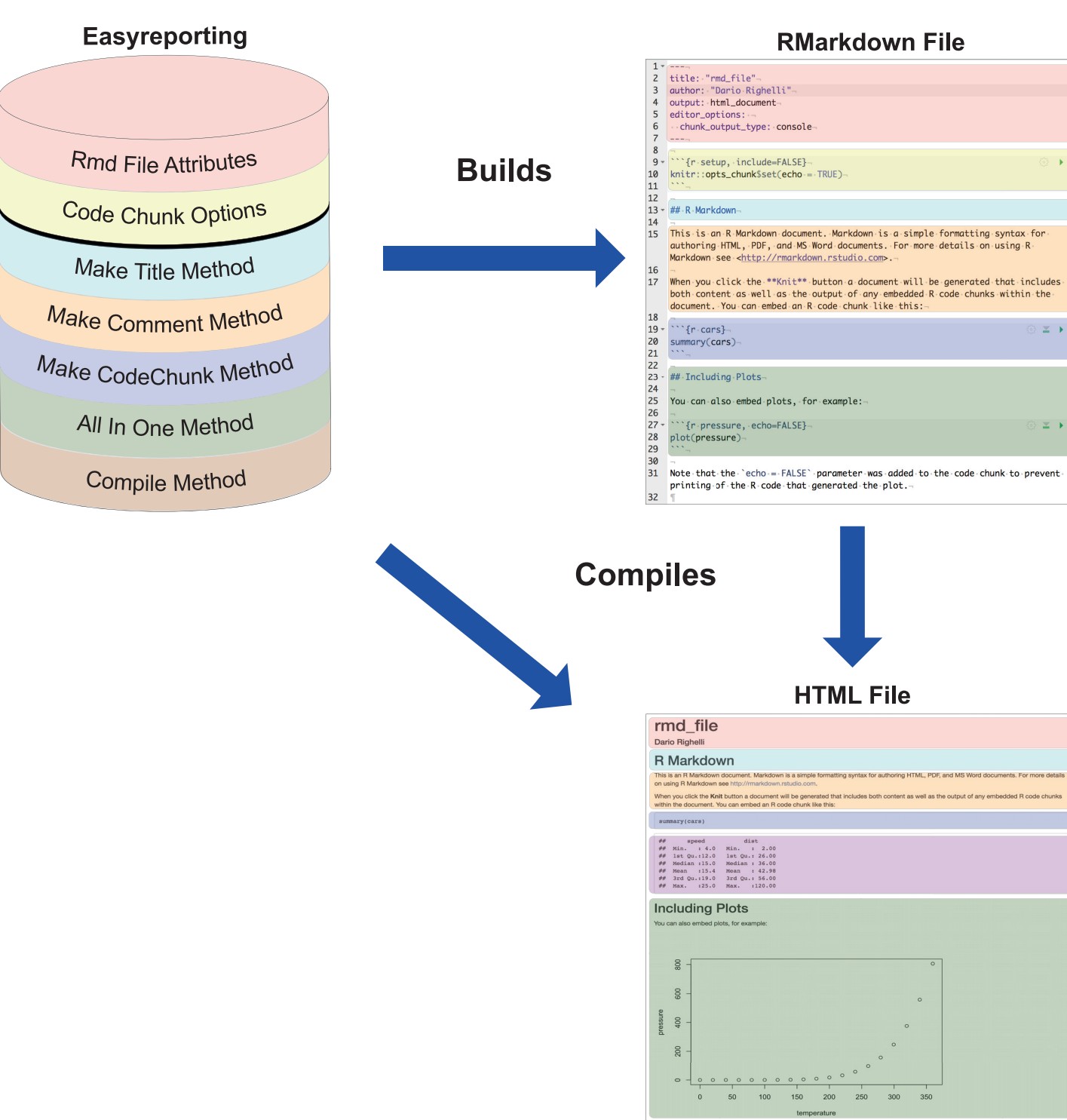

**Fig 1. The *easyreporting* class package is a representation of a *rmarkdown* file.** The color codes indicate which attribute/method represents the same-color portion of the *rmarkdown* first, and the compiled final report then (in this example an HTML file).

**Table 1. Attributes and methods of the *EasyReporting* class.**

| Attributes | Description |
| --- | --- |
| filenamePath | the report file name with the absolute path |
| title | the title of the report |
| author | the auhor |
| documentType | actually this is set to HTML |
| optionList | a list of R Markdown options |
| bibfile | a latex bibliography filename |
| resources | a data frame to store additional resources |
| **Methods** | **Description** |
| mkdTitle | Inserts an R Markdown title inside the report |
| mkdGeneralMsg | appends a general message to the report |
| mkdGeneralTitledMsg | appends a title and a general message to the report |
| mkdVariableAssignment | includes a variable assignment into the report |
| mkdCodeChunkSt | creates a CC start |
| mkdCodeChunkEnd | creates a CC end |
| mkdCodeChunkComplete | creates a complete CC |
| mkdCodeChunkCommented | creates a complete CC with a previous comment |
| mkdCodeChunkTitledCommented | creates a complete CC with a previous comment and a title |
| mkdSourceFiles | includes a list of source files inside the CC |
| compile | prints sessionInfo and compiles the R Markdown file |
| setOptionsList | set an optionList to the class |
| getOptionsList | returns the optionList from the class |
| getReportFilename | returns the report filename with path |
| getBibliography | returns the bibliography filename |
| addResource | adds en entry into the references class data frame |
| **Exported Utility Functions** | **Description** |
| makeOptionsList | makes a list of rmarkdown options that can be passed to the class |
| erGUIVolcano | executes a Shiny GUI to perform a volcano plot and trace its executed functions |

(*mkdVariableAssignement*), and finally closing the CC (*mkdCodeChunkEnd*). In this approach, it is possible to add comments with *mkdGeneralMsg* before closing the CC. ii) The second approach builds the CC in a single step by using the *mkdCodeChunkComplete* method. The method automatically embraces the tracking code into a new CC, while the user has to take care of the variable assignments and/or the function he/she wants to trace by passing it as an argument. In this approach, the user can also add personal comments passing them as an additional argument. Then, *easyreporting* automatically adds the comment before the new CC.

The first approach appears useful when one needs to carry out several R commands in a single CC. It is similar in the spirit to the functionalities offered by R-studio or other development environments, however since it is entirely command-line, it can be easily used on systems with limited development capabilities. By contrast, the second approach is more appropriate for tracing a single function call automatically. The next section will show how this second possibility can help wrap functions performing a specific step and trace their execution within GUIs or packages.

## Implementing automatic tracing functions and their usage within GUIs

The previously described CCs creation approaches can be adapted to trace several steps of an analysis pipeline and end-up with a nicely formatted and detailed analysis report. However,

they require the analysts to manually trace each step of the analysis (as he/she could also do with other available tools). Consequently, the above approaches are useful for generating analysis reports (that was the first aim of the *easyreporting* package). However, they are not suited for the automatic tracing of the steps of an analysis performed using packages function calling or point-and-click approaches through GUIs (the second aim of the *easyreporting* package).

In the last decade, GUIs are becoming very popular in bioinformatics because they simplify computational analysis allowing non-expert users to choose among several computational procedures, algorithms, and parameter settings, see for example [13, 14, 16–18]. In particular, the *shiny* (https://shiny.rstudio.com) libraries simplified the development of GUIs that incorporate the power of the statistical R language and the wide-amount of open-source packages available in repositories such as Bioconductor (https://www.bioconductor.org).

Nevertheless, computational studies obtained from GUIs might lack reproducibility since tracking all user choices is still challenging. To face this limit, the developers have to implement a RR layer when designing the GUI's back-end so that the final users can benefit from a better quality product. Moreover, the RR layer has to be transparent but understandable to not-expert users. Ideally speaking, at the end of the analysis, the user should have a human-readable report analogous to the one obtained using command-line approaches.

*Easyreporting* methods can be easily adapted to support the automatic tracing of any given function by combining a rendering function that performs the required step with a wrapping function that traces its execution. The wrapper function (WF) needs an *easyreporting* instance, and the arguments of the function to be traced (TF). Then, the developer inserts the WF in the back-end of the interface (i.e., the server if the context of a GUI is implemented with the shiny library) in the TF place. The front-end of the interface (i.e., the UI with the shiny library) remains unchanged. When the user interacts with the interface to invoke the TF, the back-end will invoke the WF, which will call both the TF function of interest and trace its usage with all parameters. In brief, employing wrapper functions makes it possible to implement a reproducible research layer within the GUI without implementing all the tracing *rmarkdown* code. Listings 4-6 illustrate a specific case with a volcano plot, and Fig 2 schematically represents the entire workflow of information.

Note that WP functions can be useful for developers or advanced users also to generate novel R packages or simply novel R-functions that automatically trace their usage. In this way, at the price of an initial effort of writing wrapper functions, their usage will be automatically traced in any context.

## Results

To better illustrate the capabilities of *easyreporting* (version 1.3.2 released with Bioconductor 3.13), we first show its usage for generating an analysis report in a case study concerning the analysis of RNA-seq data (see S1 File for details), then we illustrate how to implement a simple GUI that automatically traces the performed step with code and parameters choice to produce a report.

### Easyreporting for the creation of analysis report

The RNA-seq data used in the example allows investigating the differences in CD8+ dendritic T-cells of the immune response of two different antibodies compared with control, see [19] for more details. We chose this illustrative example since it is well-known that the analysis of RNA-seq data can lack reproducibility [20]. The dataset contains the raw counts of 37991 genes and is composed of two replicates for each of the three conditions: DEC (fd-scaDEC-205 antibody samples); E2 (E2 antibody samples) and UNTR (control samples). For illustrative

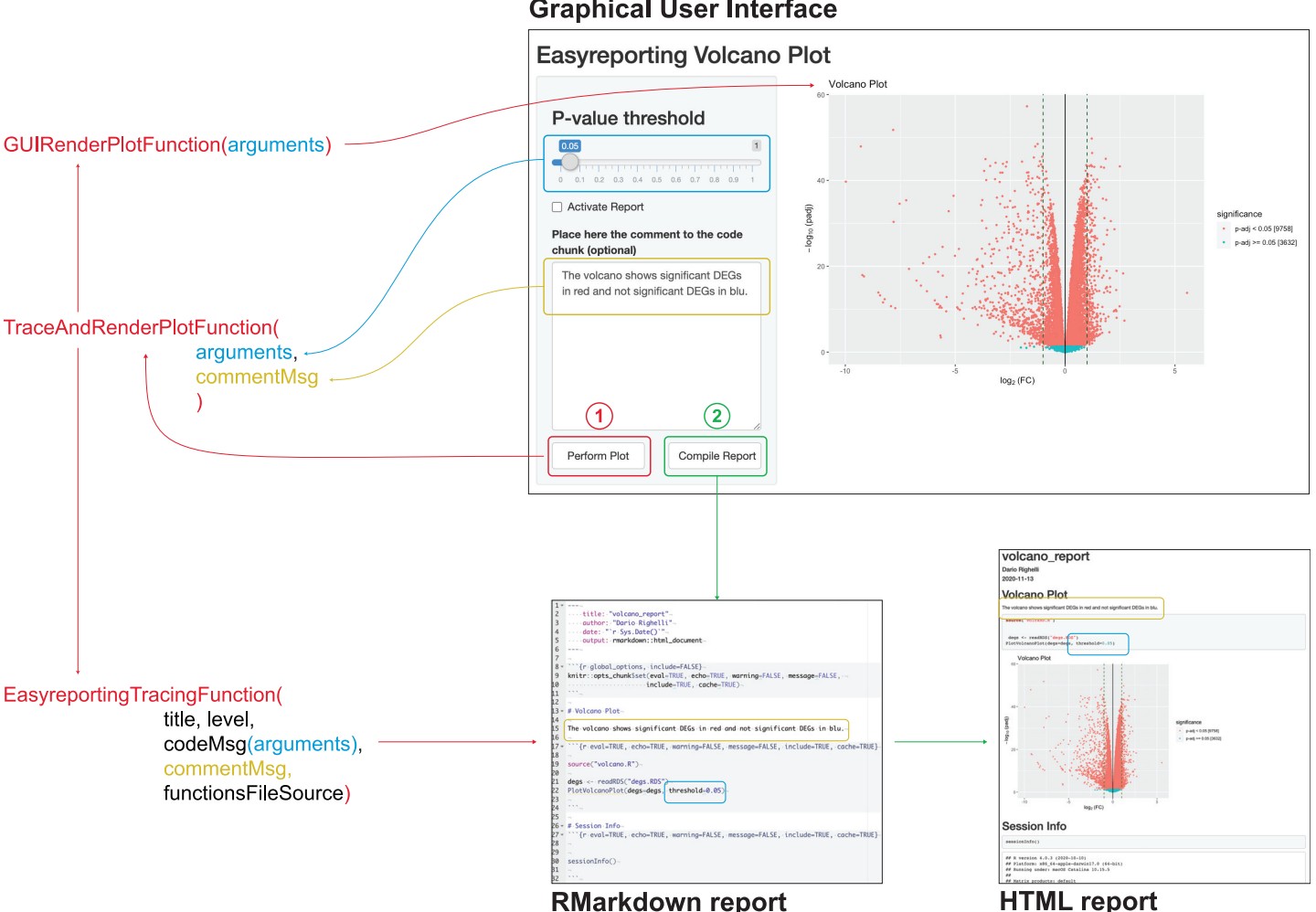

**Fig 2. Example of a graphical user interface working with *easyreporting* package.**

purposes, in our S1 File, we start the analysis from the raw count-matrix. Moreover, we released the raw counts as supplementary data with the *easyreporting* package, allowing the readers to reproduce our example. The naive pipeline will first load the data, perform some diagnostic plots, filter and normalize the raw counts, and visualize the principal component projection. It will then perform differential gene expression analysis and depict the results as a Venn diagram and MA-plots. A specific CC describes each phase.

In the following, we show the main fundamental steps that a user can adapt to any analysis, and we refer to S1 File for the detailed description of the remaining steps and S2 File for the complete report.

**Report initialization.** After loading the *easyreporting* package in the R environment, the analyst needs to initialize an analysis report by providing the file name (i.e., "rnaseq_report") and the title of the document (i.e., "RNA-seq Analysis Report"). It is also possible to specify one or more authors (i.e., "Dario Righelli") with additional associated information. For simplicity, we set-up a project directory path starting from the working directory for our report, but the user can choose other locations by setting the *filenamepath* parameter. The initialization is carried out by using the *easyreporting()* function. Note that the *filenamepath* and *title*

are mandatory parameters, while the author(s) is optional. The following Listing 1 code illustrates the initialization of a report. We refer to S1 and S2 Files (chunk of code 1) for an example where multiple authors with affiliations and additional information are provided during the initialization.

**Listing 1**. Initialization chunk

```
library("easyreporting")
  proj.path <- file.path(getwd(), "rnaseq_report")
  bioEr <- easyreporting(filenamepath = proj.path,
                title="RNA-seq Analysis Report",
                author = c("Dario Righelli"))
```

**Creation of a chunk of code.** Once the analyst has initialized the report, he/she can add a CC for each step of the analysis. As mentioned in the General Exploitation section, *easyreporting* provides two main approaches for adding CCs within a report: 1) building up the CC step by step (as shown in Listing 2) and 2) using several kinds of wrapper functions (as shown in Listing 3).

As mentioned above, in the first case, the analyst has to use the *mkdCodeChunkSt* to open a new CC. Then, he/she needs to add the code to markdown, by using the *mkdVariableAssignment* and/or the *mkdGeneralMsg* functions, for tracking variables and functions. Finally, the analyst has to close the CC using the *mkdCodeChunkEnd* function. The following Listing 2 code illustrates a step-by-step CC for loading the counts' matrix released with the package.

**Listing 2**. Step-by-step chunk construction

```
mkdTitle(bioEr, title="Loading Counts Data")
  mkdCodeChunkSt(bioEr, sourceFilesList = system.file(
      "script/importFunctions.R",
      package="easyreporting"),
      isComplete = TRUE)
  mkdVariableAssignment(bioEr, "geneCounts",
      paste("as.matrix(
      importData(system.file('",
      "extdata/BMDC_counts_FeatureCounts.xlsx', ",
      "package='easyreporting')))", sep="\n"),
      show = FALSE)
  mkdGeneralMsg(bioEr, "head(geneCounts, 20)")
  mkdCodeChunkEnd(bioEr)
```

Although the first approach leaves complete freedom to the analyst, it can be tricky for small CCs. The second approach can be more straightforward for small CCs. To this purpose, the *mkdCodeChunkComplete* function allows tracing the steps through the message parameter. The following Listing 3 code illustrates an example of a single step CC. As for the above CC, we assume that the analyst wants to read the raw counts using a user-defined function, here named "importData.R", that we stored into the importFunctions.R file available in the package "script" folder. To simplify the code writing, we embraced the calls with the *quote* function, which allows passing raw code facilitating parenthesis highlighting and function recognition to the user (see Listing 3 for an example). Additionally, it is possible to pass multiple instruction as a list of *quote* with the *c()* operator (see Section 3.3 in S1 File for an illustrative example).

**Listing 3**. One command chunk construction

```
mkdCodeChunkComplete(object = bioEr,
  code = quote(geneCounts <-
          as.matrix(importData(system.file(
```

```
                              'extdata/BMDC_counts_FeatureCounts.xlsx',
                              package='easyreporting')))),
                      sourceFilesList = system.file(
                      "script/importFunctions.R",
                      package="easyreporting"),
                      optionList = makeOptionsList(evalFlag = FALSE))
```

Note that the *mkdCodeChunkComplete* allows also to provide specific options for the CC that we are creating. In particular, in this case, turning the *evalFlag* to *FALSE* the code is not compiled during the final report construction.

It is possible to organize the report using the *mkdTitle* function. The user has to repeat this operation for each step of the analysis, as shown in S1 File. At the end of the process, it is possible to compile the *easyreporting* instance and obtain the analysis report as in S2 File in the user-defined format (default is HTML).

**Implementing automatically tracing functions.** This section shows a possible approach to encapsulate *easyreporting* methods into third-parties functions to trace the analysis step and execute the code automatically.

First of all, the developer has to write an R function that performs the analysis step of interest (such as the *MAedgeRMAPlotEx* function for rendering an MA-plot, in our example). Listings 4 shows a simple example of rendering function.

**Listing 4**. MA-plot rendering function

```
MAedgeRMAPlotEx <- function(degList)
{
  for (i in seq_along (degList)) {
  degenes <- degList [[i]] $FDR < 0.01
  with (degList [[i]], plot (logCPM, logFC,
    pch = 16, cex = 0.2, main = names (degList) [i]))
  with (degList [[i]], points (logCPM[degenes],
    logFC [degenes], col='red', pch = 16, cex = 0.2))
  }
}
```

Note that the developer does not require any extra effort at this stage. Moreover, the rendering function could also be any function available from other packages.

Then, the developer needs also to write a wrapper function (here *traceAndPlotMAPlot*). The wrapper function should take as input the arguments of the rendering function (here *MAedgeRMAPlotEx*), and a generic *easyreporting* object (here *er*). Moreover, the wrapper function has to call the *mkdCodeChunkTitledCommented* function of *easyreporting* (where we insert the rendering function call to be traced (*MAedgeRMAPlotEx*) in the *code* argument) and the call to rendering function (*MAedgeRMAPlotEx*). Listings 5 shows the wrapper function of our example.

**Listing 5**. Tracing wrapper function chunk

```
traceAndPlotMAPlot <- function(degList, er)
{
  mkdCodeChunkTitledCommented(er,
    title="Recursive Tracing Function", level = 2,
    code = quote(MAedgeRMAPlotEx(degList='degList')))
  MAedgeRMAPlotEx(degList = degList)
}
```

In this way, the wrapper function allows both to show the result and trace the function in the same step. It is easy to place the *traceAndPlotMAPlot* function-call wherever needed in the main code.

Listing code 5 shows how to use the wrapper function. In particular, we pass as input i) the object required by the rendering function (an edgeR result class in this particular example), ii) an *easyreporting* class instance (here it is the bioEr instantiated in the example).

**Listing 6**. Tracing function call chunk

```
traceAndPlotMAPlot(degList = degList, er = bioEr)
```

Note that writing the wrapper function is the only extra effort required to achieve reproducibility. This approach can be used for writing user-friendly tools, novel R-packages or simply user-defined R-functions that automatically traces their usage.

## Easyreporting for GUI implementation

To better illustrate how to incorporate a RR layer into a GUI, *easyreporting* contains a simple Shiny-App example for plotting a Volcano plot. The command *erGUIVolcano()* allows executing the app and opens the user interface. The user interface allows the user to choose a threshold for the P-value (i.e., the *P-value threshold* for detecting the significant genes in this example) and provide a text area for adding comments. In the interface, there are also two buttons (*Perform Plot* and *Compile Report*) for executing the plot and compiling the report, respectively. The *ui* function provides the code for the user interface. This code does not need to be modified to allow reproducibility. Instead, the back-end of the interface that executes the job has to incorporate a wrapper function (here *traceAndPlotVolcano*), and the call to the wrapper function, respectively. In this example, we also added the report's initialization. The *server* function provides the code for the back-end interface.

Fig 2 right side shows a schematic representation of the user interface as in the *ui* function. The left side illustrates the back-end as in the *server* function. By using the *Perform Plot* button (red box), the user activates the WF into the server-side, which in turn performs the plot and traces the executed function (the red cascade). In blue is highlighted the argument value and how it is traced through the function cascade.

Additionally, when the user adds its comments (yellow box) to the performed analysis step, the text is passed to the server.

Finally, the user can compile the report using the *Compile Report* button (green box). In this way, the server executes the *compile()* function and produces the HTML report that is automatically showed to the user.

This simple example can be generalized to complex interfaces to trace all user interactions.

## Conclusions

*Easyreporting* can be used to support RR in different analysis contexts. However, it is particularly suited for analyzing omics data and developing software/GUIs, as we have shown in this work. Compared to other previously proposed solutions such as [21], that require not negligible commitments by the final user, potentially bringing him/her to renounce to include RR inside the scripts, the implementation of a RR layer with our approach is straightforward. Moreover, it leaves maximum freedom to the developer/analyzer for automatically creating and storing an *rmarkdown* document and providing methods for its compilation and adding comments in natural language. However, in omics data analysis, it is common to use mixed scenarios and combine different programming languages depending on the availability of specific methods and functionalities in the community or to integrate analysis with queries to online databases. *Easyreporting* can provide support also for these cases, at least up to a certain level. Using interface functions between R and other programming languages, we can execute in R a few steps of an analysis implemented in other languages. For example, thanks to the R package *reticulate* it is possible to run in R some Python functions. Similarly, using suitable

API functions, we can easily interface R with on-line databases and execute the queries within the R environment. For example, using the R packages *GEOquery* or *TCGAbiolinks*, we can query and process data from the Gene Expression Omnibus (GEO) and the Genomic Data Commons (GDC) Data Portal, respectively. Nowadays, in the Bioconductor repository, many API packages allow interfacing the most common biological databases. By contrast, we should underline that if the analyses are not performed within the R environment, the user must manually document the external steps (for example, in the current release, using comments, adding.bib files, or linking to external resources as databases identifiers). Unfortunately, if the manual documentation is not detailed, the full reproducibility might be lost. Future releases of *easyreporting* will provide additional tools for handling the mixed case scenario.

Although several functionalities are already available, *easyreporting* can still benefit from some extra features such as methods for file editing, graphical representation of the analysis, and data caching. In particular, file editing can be useful for modifying specific CCs, and the graphical representation of the analysis can be useful to provide reports not only readable by third-party users but also graphically visualized as workflows. On the other hand, even though a dedicated data caching infrastructure can offer more manageability and share-ability of the data at the moment, it can be already performed in *easyreporting* by *rmarkdown* CCs option flag.

Finally, thanks to its versatility, *easyreporting* can be ideally included in any well-structured R project and the development of GUIs and packages, helping to fulfil most of the proposed rules in [2]. Moreover, if combined with virtualization solutions such as docker containers (i.e., *docker4seq* [22]) it helps to create fully reproducible projects. *Easyreporting* naturally complements docker containers in terms of reproducibility, allowing both the preserve code lines and user parameters and the computational environments and dependencies.

To conclude, our approach still requires the developer's effort to implement a RR layer into their software, which makes us imagine possible future works in this area where the code tracing is entirely left to the machine. The Java language provides a well-known example that uses the Aspect-Oriented Programming (AOP) paradigm for the software logging aspects. Unfortunately, this paradigm is still missing in the R language, but possible future approaches in the Reproducible Research area inside R could rely on implementing it, which, combined with *rmarkdown* or similar procedures, can be used to trace ad-hoc tagged functions and to log them into the report file. In such a way, reproducibility could be easier to implement and lesser subject to human errors.

## Supporting information

**S1 File. An illustrative example for the creation of an analysis report.**
(PDF)

**S2 File. The report file obtained using the analysis steps described in S1 File.**
(PDF)

## Author Contributions

**Conceptualization:** Dario Righelli, Claudia Angelini.

**Data curation:** Dario Righelli.

**Formal analysis:** Dario Righelli.

**Funding acquisition:** Claudia Angelini.

**Methodology:** Dario Righelli, Claudia Angelini.

**Software:** Dario Righelli.

**Supervision:** Claudia Angelini.

**Writing – original draft:** Dario Righelli, Claudia Angelini.

**Writing – review & editing:** Dario Righelli, Claudia Angelini.

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
