## [Decision Letter · Decision Letter 0]

3 Feb 2021

PONE-D-20-37776

Easyreporting simplifies the implementation of Reproducible Research Layers in R software

PLOS ONE

Dear Dr. Righelli,

Thank you for submitting your manuscript to PLOS ONE. After careful consideration, we feel that it has merit but does not fully meet PLOS ONE’s publication criteria as it currently stands. Therefore, we invite you to submit a revised version of the manuscript that addresses the points raised during the review process. Please take care all comments and points raised by the reviewers.

We look forward to receiving your revised manuscript.

Kind regards,

Eduardo Andrés-León

Academic Editor

PLOS ONE

Reviewers' comments:

Reviewer's Responses to Questions

**Comments to the Author**

1. Is the manuscript technically sound, and do the data support the conclusions?

Reviewer #1: Yes

Reviewer #2: Yes

2. Has the statistical analysis been performed appropriately and rigorously? 

Reviewer #1: N/A

Reviewer #2: N/A

3. Have the authors made all data underlying the findings in their manuscript fully available?

Reviewer #1: Yes

Reviewer #2: Yes

4. Is the manuscript presented in an intelligible fashion and written in standard English?

Reviewer #1: Yes

Reviewer #2: Yes

5. Review Comments to the Author

Reviewer #1: I have been reading your manuscript, as well as have had a look at `easyreporting` code. As you have already stated, this R package is focused on human-focused reports generation which help on research reproducibility. So, which strategy would you recommend to generate reports which involve mixed scenarios, like running a first part of R code which generates both intermediate and final results, a second part which uses these results with one or more web services or third party tools, getting additional results, and a last part of R code which uses as input a selection of the previous result files? I guess you could generate a report for the R dependent parts, but I'm missing your recommendation for this kind of scenario in the manuscript or in supplementary materials. I'm also missing some clearer hints to the readers of your manuscript related to provenance tracking or provenance injection in the generated report which help to obtain all the supplementary data needed to successfully reproduce the analysis.

Page 3, paragraph starting at line 73. I agree that it is relevant for the end user realizing that an analysis report is an instance of `easyreporting`, but is relevant for the end user of the package to learn the technical detail of `easyreporting` being structured as an S4 class (http://adv-r.had.co.nz/S4.html)?

I have also been having a look at the supplementary materials 1, and its associated repository https://github.com/drighelli/easyreporting_supplementary , and I have found an issue against reproducibility outside the installation where the report was generated. The file `rnaseq_report_live.Rmd` at `Report_files` directory, generated from `report.R` (on the same directory), has next sentence on each of its codeblocks

source("/Library/Frameworks/R.framework/Versions/4.0/Resources/library/easyreporting/script/importFunctions.R")

(permalink to first occurrence https://github.com/drighelli/easyreporting_supplementary/blob/ef9e498d4eb431dd488cc7db615a4c3fe3d19e1f/Report_files/rnaseq_report_live.Rmd#L15 )

That sentence has an absolute path, derived from

mkdCodeChunkSt(bioEr, sourceFilesList=system.file("script/importFunctions.R", package="easyreporting"), isComplete=TRUE)

(permalink https://github.com/drighelli/easyreporting_supplementary/blob/ef9e498d4eb431dd488cc7db615a4c3fe3d19e1f/Report_files/report.R#L11 )

so it limits the reproducibility, as codeblocks cannot be run as is. It would not work in a Linux or a Windows installation without a extensive revision, for instance. Also, the report could disclose sensitive data related to these absolute paths.

I have several recommendation for future releases of `easyreporting`. In analyses with more than one author, package should allow showing all of them on the generated reports. I'm also missing richer ways to attach metadata related to an specific analysis:

* `easyreporting` should provide a mechanism to attach the optional ORCID, Researcher Id or similar of each author, in order to avoid ambiguity among researchers with similar names.

* The report could optionally contain a list of used files in the analysis, both sourced scripts and input files, along with their digest. This could help identifying whether the report is stale because either the scripts or input files have changed.

* I'm missing some specific method which allows injecting high level annotations about the inputs. If some analysis depends on specific public or under request data which are not provided by R / Bioconductor packages, there should be a standardized or, at least, recommended way to tell where to find those inputs, or publicly accepted identifiers associated to the data (i.e. EGA or COSMIC ids, DOIs, etc...). These annotations in the generated report would help a lot when some researcher tries to reproduce the described analysis, but it depends on the report authors' collaboration.

Reviewer #2: The authors present a new R package to facilitate the creation of reproducible research reports. Their package can be added to a growing number of resources for this important task and it's commendable that researches from within the *omics are investing time and ingenuity to improve the mechanics of the modern, largely computational scientific process in these fields.

So while this is by itself a great goal, the paper did not convince me that the suggested workflow for reporting script based analysis is superior to using simple RMarkdown reports. RMarkdown is not (!) a complicated markup language and learning it appears to me less involved then using the easyreporting wrapper functions, which add another layer of complexity to the already long RMarkdown rendering pipeline. So I would suggest to the authors not to focus on this first (rather theoretical) usecase of easyreporting, but to go all in on the second one, which is the ability to add a reporting level to Shiny applications. Here the paper presents indeed a novel solution which might be relevant AND practical. Of course the paper then only addresses the admittedly small group of R Shiny developers, but again I do not see how normal R users outside of this circle would benefit from the suggested additional layer on top of RMarkdown.

So accordingly I suggest to change the title, wording and section order of the paper to emphasise the Shiny-enhancing feature of the package. Here I would also ask the authors to state more clearly that this is only intended to work with Shiny apps, not any GUI framework. Shiny is of course only one of many GUI systems and it should be clear from the beginning that easyreporting can only provide a human-readable logging system for this specific one. Shiny is simple and popular at the moment, but also too slow and fickle for many of the large-data applications in the *omics.

These were my main comments. Finally a list of minor observations:

- I think it would be better to speak more generally about "virtualisation solutions such as Docker containers" instead of "dockers". The authors should add a reference to a paper explaining these solutions. I never read the term "dockers" for "Docker containers".

- As the authors present a new R package they should make it clear in the introduction which package version the paper describes exactly.

- The text in the subsection "easyreporting for GUI implementation" is a duplication of the text in the caption for figure 2. The figure caption can be shortened.

- The code in strings for mkdCodeChkunkSt, mkdVariableAssignment, etc. is not a good solution, because it makes writing the respective code tricky (no syntax highlighting etc.). Maybe the authors could come up with a better implementation based on base::quote() or base::deparse().

b <- quote({

a <- 1 + 2

a + a

})

format(b)

eval(b)

I am not qualified to review the scientific validity of the case study presented in the supplementary material.

6. PLOS authors have the option to publish the peer review history of their article (what does this mean?). If published, this will include your full peer review and any attached files.

Reviewer #1: **Yes: **José M. Fernández

Reviewer #2: No

---

## [Author Response · Author response to Decision Letter 0]

22 Mar 2021

Dear Editor and Reviewers,

We are submitting a revised version of our manuscript, "Easyreporting simplifies the implementation of Reproducible Research Layers in R software," which addresses most of the reviewers' issues.

We thank the reviewers for their suggestions that allowed us to improve the paper and the related software. We believe that the revised version of our work can be considered for publication in PLOS ONE.

The point-by-point answers to the Reviewers questions are listed in the following paragraph.

Reviewer #1: 

Q1- I have been reading your manuscript, as well as have had a look at `easyreporting` code. As you have already stated, this R package is focused on human-focused reports generation which help on research reproducibility. So, which strategy would you recommend to generate reports which involve mixed scenarios, like running a first part of R code which generates both intermediate and final results, 

a second part which uses these results with one or more web services or third party tools, getting additional results, and a last part of R code which uses as input a selection of the previous result files? 

I guess you could generate a report for the R dependent parts, but I'm missing your recommendation for this kind of scenario in the manuscript or in supplementary materials. 

I'm also missing some clearer hints to the readers of your manuscript related to provenance tracking or provenance injection in the generated report which help to obtain all the supplementary data needed to successfully reproduce the analysis.

R1 - We thank the referee for the interesting comments. Easyreporting is an R package. Therefore, it allows supporting reproducible research when the analyses are performed within the R environment. In the case of a mixed scenario to preserve full reproducibility is more challenging. In Section Conclusion, we added a discussion about how to preserve reproducibility in such cases and when the reproducibility might be lost. 

“However, in omics data analysis, it is common to use mixed scenarios and combine different programming languages depending on the availability of specific methods and functionalities in the community or to integrate analysis with queries to on-line databases. \\textit{Easyreporting} can provide support also for these cases, at least up to a certain level. Using interface functions between R and other programming languages, we can execute under R a few steps of an analysis implemented in other languages. For example, thanks to the R package \\textit{reticulate} it is possible to run under R some python functions. Similarly, using suitable API functions, we can easily interface R with on-line databases and execute the queries within the R environment. For example, using the R packages \\textit{GEOquery} or \\textit{TCGAbiolinks}, we can query and process data from the Gene Expression Omnibus (GEO) and the Genomic Data Commons (GDC) Data Portal, respectively. Nowadays, in the Bioconductor repository, many API packages allow interfacing the most common biological databases. By contrast, we should underline that if the analyses are not performed within the R environment, the user must manually document the external steps (for example, in the current release, using comments, adding .bib files, or linking to external resources as databases identifiers). Unfortunately, if the manual documentation is not detailed, the full reproducibility might be lost. Future releases of \\textit{easyreporting} will provide additional tools for handling the mixed case scenario.” 

Q2- Page 3, paragraph starting at line 73. I agree that it is relevant for the end user realizing that an analysis report is an instance of `easyreporting`, but is relevant for the end user of the package to learn the technical detail of `easyreporting` being structured as an S4 class (http://adv-r.had.co.nz/S4.html)?

R2 - The class provided is a schematic representation of an rmarkdown file. For end-users that want to create a report it is not relevant knowing that it is a S4 class. However, for end-users that are developers and want to use `easyreporting` in other products, it is important, since the S4 class is treated differently from R5 or R6 classes.

Q3. I have also been having a look at the supplementary materials 1, and its associated repository https://github.com/drighelli/easyreporting_supplementary , and I have found an issue against reproducibility outside the installation where the report was generated. The file `rnaseq_report_live.Rmd` at `Report_files` directory, generated from `report.R` (on the same directory), has next sentence on each of its codeblocks

source("/Library/Frameworks/R.framework/Versions/4.0/Resources/library/easyreporting/script/importFunctions.R")

(permalink to first occurrence https://github.com/drighelli/easyreporting_supplementary/blob/ef9e498d4eb431dd488cc7db615a4c3fe3d19e1f/Report_files/rnaseq_report_live.Rmd#L15 )

That sentence has an absolute path, derived from

mkdCodeChunkSt(bioEr, sourceFilesList=system.file("script/importFunctions.R", package="easyreporting"), isComplete=TRUE)

(permalink https://github.com/drighelli/easyreporting_supplementary/blob/ef9e498d4eb431dd488cc7db615a4c3fe3d19e1f/Report_files/report.R#L11 )

so it limits the reproducibility, as code blocks cannot be run as is. It would not work in a Linux or a Windows installation without an extensive revision, for instance. Also, the report could disclose sensitive data related to these absolute paths.

R3 - We thank the reviewer for raising this issue in our original approach. We improved our external source script depending methods by automating the copy of the imported file scripts to solve this problem locally. This is reflected on the final supplementary analysis report where the absolute paths in the source calls are substituted simply by the script name (which has previously been copied in the same folder of the report).

Q4- I have several recommendation for future releases of `easyreporting`. In analyses with more than one author, package should allow showing all of them on the generated reports. I'm also missing richer ways to attach metadata related to an specific analysis: `easyreporting` should provide a mechanism to attach the optional ORCID, Researcher Id or similar of each author, in order to avoid ambiguity among researchers with similar names.

R4 - We thank the reviewer for these recommendations. 

Following these suggestions, since version 1.3.2 we implemented the possibility to add more than one author accompanying each of them with an email, ORCID, affiliation, affiliation website, personal url. 

We added this information in Section “General Description and Initialization”, and a practical example is in the first chunk of code of Supplementary File 1.

We also update Table 1 with the new attributes.

Q5 - The report could optionally contain a list of used files in the analysis, both sourced scripts and input files, along with their digest. This could help identifying whether the report is stale because either the scripts or input files have changed.

R5 - As described at point 3, we copy each used script in the report directory.

Q6 - I'm missing some specific method which allows injecting high level annotations about the inputs. If some analysis depends on specific public or under request data which are not provided by R / Bioconductor packages, there should be a standardized or, at least, recommended way to tell where to find those inputs, or publicly accepted identifiers associated to the data (i.e. EGA or COSMIC ids, DOIs, etc...). These annotations in the generated report would help a lot when some researcher tries to reproduce the described analysis, but it depends on the report authors' collaboration.

R6 - We thank the referee for this suggestion. 

Since version 1.3.2 on Bioconductor 3.13, we provided two additional methods for adding additional resources to the final report.

Please note that Bioconductor 3.13 is on the official Bioconductor at the moment in the development state and will be released as stable from the next beginning of April.

The first one is based on a bibliography (.bib) file that needs to be specified during the report creation, which cited articles will be included in the final report during its compilation.

We added this information in the “General description and Initialization” Section:

“Additionally, during the class creation, it is possible to define a bibliography latex file through the \\textit{bibliography} argument, that will be compiled with the report (see table \\ref{tab1} for additional details).”

The second one a more user-free approach where the user can provide additional resources in the form of source-reference-description with the method addResource().

This method will collect the resources in a data.frame that will be added in a specific section named “Resources Availability” in the final report during the compilation.

We referred to this approach in the “General Description and Implementation” Section:

“Additionally, this process creates two others optional sections, one with the cited references when a bibliography file has been specified during the easyreporting instance creation, and another named \\textit{Resources Availability} with the resources specified with the \\textit{addResource} method (see Supplementary Section 3.9 for more detail). “

We update Table 1 with the new attributes and functionalities.

Reviewer #2: 

The authors present a new R package to facilitate the creation of reproducible research reports. Their package can be added to a growing number of resources for this important task and it's commendable that researches from within the *omics are investing time and ingenuity to improve the mechanics of the modern, largely computational scientific process in these fields.

So while this is by itself a great goal, the paper did not convince me that the suggested workflow for reporting script based analysis is superior to using simple RMarkdown reports. RMarkdown is not (!) a complicated markup language and learning it appears to me less involved then using the easyreporting wrapper functions, which add another layer of complexity to the already long RMarkdown rendering pipeline. 

Q1- So I would suggest to the authors not to focus on this first (rather theoretical) usecase of easyreporting, but to go all in on the second one, which is the ability to add a reporting level to Shiny applications. Here the paper presents indeed a novel solution which might be relevant AND practical. Of course the paper then only addresses the admittedly small group of R Shiny developers, but again I do not see how normal R users outside of this circle would benefit from the suggested additional layer on top of RMarkdown.

R1 - We thank the reviewer for this comment. We agree that the main advantage of "easyreporting" is to support developers in implementing RR layers in other tools such as GUIs or other R-packages. Despite that, we do not think that the example of using "easyreporting" for writing an analysis report should be removed, and the use of "easyreporting" should be limited to the development of shiny interfaces. 

We agree with the referee that if the user is only interested in writing analysis reports, there might be other alternatives to "easyreporting". We also agree that learning rmarkdown is not a problem. Therefore, we relaxed the presentation of "easyreporting" in such a context, and we make a more clear statement about the advantages of "easyreporting" for developers.

In this context, we believe that "easyreporting" can be used not only inside Shiny applications (as shown in our example), but in principle also in any general Graphical User Interface. Moreover, it can also be used in any R package or any user-defined R-script. We modified the manuscript to make this statement more clear.

Q2 - So accordingly I suggest to change the title, wording and section order of the paper to emphasise the Shiny-enhancing feature of the package. 

R2 - As mentioned in the previous point, we tried to make this point more clear, without changing the structure of the manuscript. 

Q3 - Here I would also ask the authors to state more clearly that this is only intended to work with Shiny apps, not any GUI framework. Shiny is of course only one of many GUI systems and it should be clear from the beginning that easyreporting can only provide a human-readable logging system for this specific one. Shiny is simple and popular at the moment, but also too slow and fickle for many of the large-data applications in the *omics.

R3 - As mentioned in point 1, we do not think that our software is necessarily related to the Shiny applications. Indeed any GUI framework allows the implementation of user input fields like the one we showed in our Shiny example. Easyreporting can work with any interface with the core of the analysis executed in the R environment. 

We agree that the Shiny libraries are very popular at the moment within the R community, although they have some limits. There are several shiny applications developed for omics data analysis and visualization. Some examples are the iSEE Bioconductor package for single-cell data visualization, the HiceekR package for the Hi-C omics data analysis and visualization, and many others. The efficiency of the code is often related to the R implementation rather than to the shiny interface. However, R can be interfaced with C++ and well as python to make more efficient codes.

Q4- These were my main comments. Finally a list of minor observations:

I think it would be better to speak more generally about "virtualisation solutions such as Docker containers" instead of "dockers". The authors should add a reference to a paper explaining these solutions. I never read the term "dockers" for "Docker containers".

R4 - Thanks for underlining such a point. We corrected the term through the manuscript.

Q5 - As the authors present a new R package they should make it clear in the introduction which package version the paper describes exactly.

R5 - Thanks for reporting this gap, we reported this information in the Results Section:

“To better illustrate the capabilities of \\textit{easyreporting} (version 1.3.2 released with Bioconductor 3.13)“

Bioconductor 3.13 is on the official Bioconductor at the moment in the development state and will be released as stable from the next beginning of April.

Q6 - The text in the subsection "easyreporting for GUI implementation" is a duplication of the text in the caption for figure 2. The figure caption can be shortened.

R6 - Thanks we removed the duplicated part.

Q7 - The code in strings for mkdCodeChkunkSt, mkdVariableAssignment, etc. is not a good solution, because it makes writing the respective code tricky (no syntax highlighting etc.). Maybe the authors could come up with a better implementation based on base::quote() or base::deparse().

R7 - We thank the reviewer for this useful suggestion. 

We followed these guidelines for allowing the insertion of code through the quote() function inside our functions.

We also provided the possibility to parse more than one instruction at the same time when the user provides a list of quote such as 

c(quote(instruction1), quote(instruction2))

We reported this improvement in the “Creation of a chunk code” Section:

“To simplify the code writing, we embraced the calls with the \\textit{quote} function, which allows to pass raw code facilitating parenthesis highlighting and function recognition to the user (see Listing \\ref{verb3} for an example).

Additionally, it is possible to pass multiple instruction as a list of \\textit{quote} with the \\textit{c()} operator (see Supplementary File 1 Section 3.3 for an illustrative example)”

---

## [Decision Letter · Decision Letter 1]

21 Apr 2021

Easyreporting simplifies the implementation of Reproducible Research Layers in R software

PONE-D-20-37776R1

Dear Dr. Righelli,

We’re pleased to inform you that your manuscript has been judged scientifically suitable for publication and will be formally accepted for publication once it meets all outstanding technical requirements.

Kind regards,

Eduardo Andrés-León

Academic Editor

PLOS ONE

Additional Editor Comments (optional):

Please take into account the comments regarding the R versions

Reviewers' comments:

Reviewer's Responses to Questions

**Comments to the Author**

1. If the authors have adequately addressed your comments raised in a previous round of review and you feel that this manuscript is now acceptable for publication, you may indicate that here to bypass the “Comments to the Author” section, enter your conflict of interest statement in the “Confidential to Editor” section, and submit your "Accept" recommendation.

Reviewer #1: All comments have been addressed

Reviewer #2: All comments have been addressed

2. Is the manuscript technically sound, and do the data support the conclusions?

Reviewer #1: Yes

Reviewer #2: (No Response)

3. Has the statistical analysis been performed appropriately and rigorously? 

Reviewer #1: N/A

Reviewer #2: (No Response)

4. Have the authors made all data underlying the findings in their manuscript fully available?

Reviewer #1: Yes

Reviewer #2: (No Response)

5. Is the manuscript presented in an intelligible fashion and written in standard English?

Reviewer #1: Yes

Reviewer #2: (No Response)

6. Review Comments to the Author

Reviewer #1: I have read the revised manuscript and supplementary materials, as well as the solutions and answers to each one of the issues from the referees. When I read that easyreporting release 3.12 will appear on ongoing Bioconductor 3.13, I went to https://bioconductor.org/developers/release-schedule/ , and I realized Bioconductor 3.13 will be tied to R-4.1.0, which will be released maybe on 2021-05-18. So, I started wondering how backward compatible are usually bioconductor packages, in terms of R language and dependences on other package.

So, my question to you is, how backward compatible is easyreporting? Could release 1.3.2 of easyreporting be used in R 4.0.x or 3.6.x , for instance? File https://github.com/drighelli/easyreporting/blob/master/DESCRIPTION is not declaring any minimal version of R, but for instance one of its imports, rlang, depends at least on R >= 3.3.0

Page 6, line 174: I guess there is an unescaped LaTeX underscore in the URL, as it should be https://github.com/drighelli/easyreporting_supplementary

Reviewer #2: (No Response)

7. PLOS authors have the option to publish the peer review history of their article (what does this mean?). If published, this will include your full peer review and any attached files.

Reviewer #1: **Yes: **José M. Fernández

Reviewer #2: No

---

## [Editor Report · Acceptance letter]

28 Apr 2021

PONE-D-20-37776R1 

Easyreporting simplifies the implementation of Reproducible Research Layers in R software 

Dear Dr. Righelli:

I'm pleased to inform you that your manuscript has been deemed suitable for publication in PLOS ONE. Congratulations! Your manuscript is now with our production department. 

Kind regards, 

on behalf of

Dr. Eduardo Andrés-León 

Academic Editor

PLOS ONE